# Hardware functional obfuscation with ferroelectric active interconnects

Tongguang Yu[1], Yixin Xu [1], Shan Deng[2], Zijian Zhao[2], Nicolas Jao[1], You Sung Kim[3], Stefan Duenkel[3], Sven Beyer[3], Kai Ni [2✉], Sumitha George [4✉] & Vijaykrishnan Narayanan[1]

Existing circuit camouflaging techniques to prevent reverse engineering increase circuit-complexity with significant area, energy, and delay penalty. In this paper, we propose an efficient hardware encryption technique with minimal complexity and overheads based on ferroelectric field-effect transistor (FeFET) active interconnects. By utilizing the threshold voltage programmability of the FeFETs, run-time reconfigurable inverter-buffer logic, utilizing two FeFETs and an inverter, is enabled. Judicious placement of the proposed logic makes it act as a hardware encryption key and enable encoding and decoding of the functional output without affecting the critical path timing delay. Additionally, a peripheral programming scheme for reconfigurable logic by reusing the existing scan chain logic is proposed, obviating the need for specialized programming logic and circuitry for keybit distribution. Our analysis shows an average encryption probability of 97.43% with an increase of 2.24%/ 3.67% delay for the most critical path/ sum of 100 critical paths delay for ISCAS85 benchmarks.

[1] Pennsylvania State University, State College, PA 16802, USA. [2] Rochester Institute of Technology, Rochester, NY 14623, USA. [3] GlobalFoundries Fab1 LLC & Co. KG, Dresden, Germany. [4] North Dakota State University, Fargo, ND 58102, USA. ✉email: kai.ni@rit.edu; sumitha.george@ndsu.edu

Hardware security is becoming increasingly prominent with globalization and outsourcing integrated circuit (IC) fabrication to various foundries[1]. A major threat to hardware security is reverse engineering (RE). Objects ranging from large aircraft to the smallest microchips are vulnerable to RE[2]. Attackers' motives may include commercial piracy, intelligence, patent laws[3]. RE techniques can enable the attacker to inject a hardware Trojan, copy propriety IPs, extract hard-coded keys, and copy instruction sequences[4]. Such scenarios necessitate the need for hardware encryption in chips, which adds a level of difficulty to IC analysis[5] and RE. RE extracts information from an IC utilizing techniques like depackaging, delayering, high resolution imaging and side-channel probing, etc[3]. For example, attackers often depackage the target chip, take high definition image of each layer, and then use an image recognition software to extract netlists[4]. Different layout shapes of different logic cells make this process easy for the attackers to gather logic information. To mitigate such risks, an effective technique is to add camouflaged cells in the design such that discerning logic process through RE is difficult or impossible. Camouflaged cells prevent the interpretation of correct functionality by being logically obscured.

Various gate-level camouflaging techniques have been developed with conventional CMOS devices[6–10]. Traditional CMOS-based camouflaging implementations incur overheads in circuit area, power, and delay. Recent explorations have investigated emerging devices such as spin-transfer-torque devices[11], tunnel-FETs[12], ferroelectric devices[13,14], tungsten diselenide (WSe$_2$) devices[15], etc., for provisioning hardware security by leveraging unique properties such as their non-volatile behavior. Rajendran et al.[16] proposed a gate camouflaging technique by inserting dummy via/contacts in the layout and creating look-alike layouts for NAND, NOR, and XNOR cells (Fig. 1a). With layout look-alike camouflaged gates, the attackers may interpret the function incorrectly and end up with a faulty netlist. However, advances in imaging and computer vision technology have made such methods less effective and susceptible to direct probing attacks[6]. One approach to avoid probing attacks is to use internal parameters of devices, such as product invariability or different states of the devices, for the implementation of different functions while retaining identical physical layouts[6,17]. Wu et al.[14] proposed that two-dimensional black phosphorus field-effect transistors with reconfigurable polarities are suitable for hardware security applications (Fig. 1a). These transistors can be dynamically switched between p-FET and n-FET operations through electrostatic gating. Though this approach achieves minimum area overhead, its integration with Si CMOS technology is challenging.

Erbagci et al.[6] proposed a gate camouflaging technique using threshold voltage defined (TVD) logic topology. The key idea relies on the usage of different threshold voltage ($V_{TH}$) transistors but with identical physical layouts. Their work introduced a generic 2-input TVD logic gate capable of realizing multiple logic functions (NAND, NOR, and XNOR). They achieved this by setting pull-down transistors with different $V_{TH}$ implantations (i.e., low-$V_{TH}$ (LVT), and high-$V_{TH}$ (HVT)). However, this circuit does not provide flexible reconfigurability as the $V_{TH}$ of conventional CMOS transistors are not run-time programmable. Dutta et al.[17] further enhanced the TVD device design (Fig. 1a) by replacing the pull-down logic transistors with emerging ferroelectric FETs (FeFETs). By utilizing the feature of voltage-dependent polarization switching of FeFET, pull-down transistors can be reprogrammed into LVT and HVT states. Exploiting the programmable $V_{TH}$ of the FeFETs makes the TVD logic gate-level camouflaging and run-time reconfigurable simultaneously. However, these features come at the expense of complex design with differential logic, high area, power, timing expense.

This work proposes a simple area efficient reconfigurable logic which can act as encryption key logic. To avoid IC counterfeiting, the functional IP is locked with key logic and the IP can be unlocked with a correct sequence of keys given to the trusted customer. Therefore protection can be achieved by intentionally programming the device with incorrect keys. Our proposed scheme for securing the ICs by hardware encryption is shown in Fig. 1b. In this scheme, an active interconnect based encryption block is designed and is chosen to judiciously place them at different locations in the chip. Keybits are used to program encryption logic. An example arrangement of encryption blocks C1, C2, C3 and C4 in an IC is shown in Fig. 1b. In this scheme, not all gates need to be camouflaged. Rajendran et al.[16] has shown that choosing a subset of gates to be camouflaged is sufficient to make the IC immune to RE. One advantage with this simple yet but powerful technique is that, the placement of key logic can be in the non-critical timing branch of the logic, yet the output function will be encrypted. Further analysis shows that the strategic placement of the key circuit influences the output without posing challenges in the timing closure. In this way, the proposed scheme causes only minimal interference to the actual circuit in terms of delay, area and power.

It is crucial to develop an encryption logic with efficient functional implementation, resistance to hardware attacks, CMOS compatibility, high density and minimal overheads. The ability to program the encryption block multiple times during run-time is significant for enhanced security. To satisfy aforementioned requirements, a compact encryption block is designed as shown in Fig. 1c. Here the block needs to be developed in such a way that the output signal gets inverted or non-inverted based on the internal state of the switch. The switch can be either in closed or in open state. Note that complementary states are maintained in the upper and lower branches. If the switch in the upper branch is in a conducting (closed) state then inverted input appears in the output and if the switch in the lower branch is in a conducting (closed) state then the output follows input. Thus with reconfigurability, the proposed encryption block can act as either an inverter or buffer. The vision is to integrate the reconfigurability to switch's internal state such that physical layout looks identical in both modes of operation. Hence the block can act as a camouflaging buffer-inverter standalone gate. In addition, the usage of this block in conjunction with other complex gates by placing it at their input or output, the overall functionality changes, extending the camouflage to complex circuits.

There are many promising memory technologies available to realize the switch in the reconfigurable encryption block, each with its own unique features. Figure 1c shows the potential implementations of the switch using SRAM, Flash, resistive RAM (ReRAM), phase change memory (PCM), spin transfer torque magnetic RAM (STT-MRAM), and FeFETs. SRAM is the most straightforward memory to use but is volatile and typically requires at least six transistors, dissipating significant leakage power while suffering from low memory density. A Flash memory-based switch is non-volatile and compact[18], but memory programming is slow (~ms) and requires a high programming voltage (~10 volts). In addition, it is challenging to scale the embedded flash memory to 28 nm and below due to the thick gate stack and also added costs from the additional masks with scaling[19]. Therefore, for embedded flash memory to use as a compact non-volatile switch, significant challenges remain. Emerging non-volatile memory (NVM)-based switches have also been proposed and superior performance has been demonstrated[20–22]. Resistive memories are a class of two terminal NVM devices, including ReRAM, PCM, and STT-MRAM. Information is stored as conductive filament formation or rupture (ReRAM), film crystallization or amorphization (PCM), or

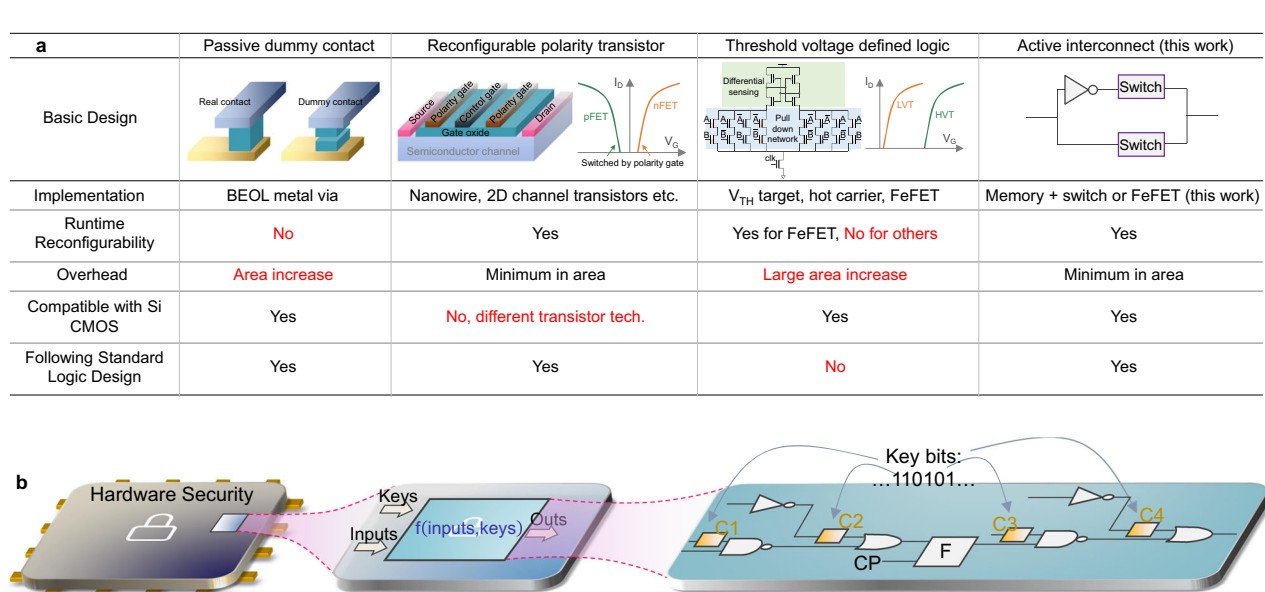

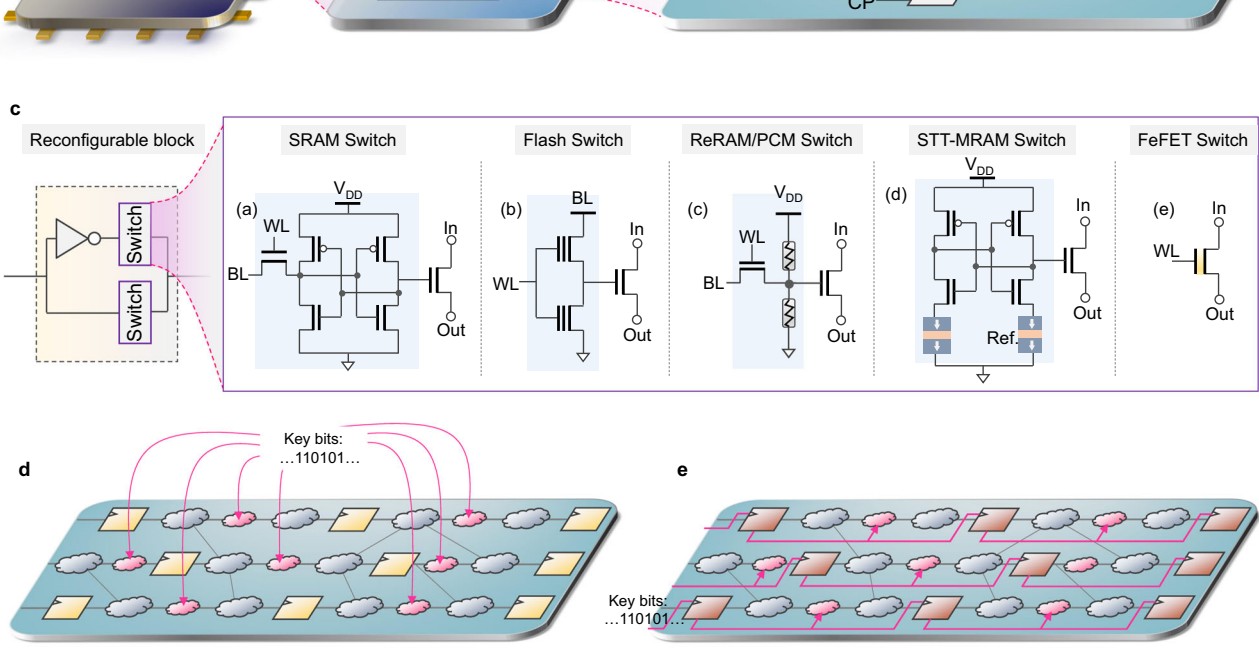

**Fig. 1 Overview of the proposed encryption of IC logic design harnessing the ultra-compact FeFET active interconnect reconfigurable switch.**
**a** Comparison between different camouflage logic designs. The proposed active interconnect based approach is advantageous in realizing the camouflage logic. **b** Illustration of utilizing the active interconnect based encryption block for obfuscating the IC logic. The keys are to dynamically reconfigure the interconnect such that the logic function is hidden. **c** The encryption block and implementations of switches using various technologies to realize camouflaging logic function. FeFET realizes a reconfigurable ultra-compact active interconnect switch. **d** Distribution challenge of keybits to the programmable encryption logic block. **e** Distribution of keybits to the programmable encryption logic block via scan chain logic.

parallel or anti-parallel orientation of the magnetization in a magnetic tunnel junction (STT-MRAM). These devices are non-volatile and compact, but usually require a large conduction current to program the devices, consuming a significant write power. The limited on/off resistance ratio (~100 for ReRAM/PCM and ~5 for STT-MRAM) usually requires additional circuitry, such as the 1T2R structure in ReRAM/PCM[20,22] and an even more complex supporting structure for STT-MRAM[21] to realize a non-volatile switch.

In this work, a non-volatile active interconnect switch, based on a single-FeFET is proposed to build the reconfigurable encryption block. In a FeFET, the ferroelectric layer is integrated as the gate dielectric of a MOSFET, where the information is stored in the direction of the ferroelectric polarization, which can be switched with an applied electric field. By configuring the direction of the polarization to point toward the semiconductor channel or the gate electrode, the device is set to either low-$V_{TH}$ or high-$V_{TH}$ state respectively. This makes FeFET an integrated single transistor memory, a great advantage to realize the non-volatile switch. The dynamic reconfigurability of $V_{TH}$ state has been harnessed in many applications, for instance, on memory-centric computing[17,23–28]. Since the ferroelectric memory is written with an electric field rather than a large conduction current[26], this technology becomes highly energy efficient

(e.g., down to ~1 fJ/bit write energy). Therefore the dynamic reconfigurability of $V_{TH}$, along with its intrinsic three-terminal structure, nonvolatility, superior write performance, excellent CMOS compatibility, and scalability[29,30], shape FeFET a prime candidate for the non-volatile switch. Demonstrations of FeFET on advanced transistor technologies, such as the 22 nm fully-depleted silicon-on-insulator[31], FinFET[32] and gate-all-around transistor[33,34], have been reported, demonstrating the great promise of scaling for FeFET. The three terminal device structure makes the FeFET a very compact active interconnect. The key idea behind our proposed active interconnect based reconfigurable encoding circuit leverages the run time reconfigurability of the encryption block by manipulating the threshold voltage of FeFET[35–38].

Note that design variants of our proposed active interconnect based dynamically configurable block can be extended to offer various chip design applications. For example, an active configurable route switching can be enabled, as shown in Fig. S1, to route a signal to different functional units. The directions can be tuned with programming configuration. Another example is a configurable path connector that connects/disconnects inputs to destination units. This is especially beneficial for controlling the logic toward redundant functional units. Redundant functional units are typically used in chips as means to increase the reliability against fault tolerance. Reconfigurable logic gate is another byproduct of the proposed method which is realizable by programming the control inputs gates. Many combinations such as inverter, NAND, AND, OR, NOR, XOR and XNOR are possible (Fig. S1). In addition, reconfigurable gates can be deployed in chips to tackle Engineering Change Orders (ECO)[39,40], where functional logic changes need to be met with minimal layout changes. The ability to meet functional changes with existing gates in the design is relevant for both pre-mask and post-mask ECOs.

All dynamic logic programming schemes including the aforementioned dynamic encryption programming pose a challenge in getting the desired input values to the configurable logic which mostly requires appropriate write voltages to set its internal state. This necessitates a robust peripheral logic and circuitry. However, a systematic approach for peripheral programming has rarely been explored in recent reconfigurable logic research works. Nowadays application specific integrated circuit(ASIC)/system on a chip (SoC) implementations come with more than a million gates and flipflops spread across the entire chip. As the amount of logic increases, the number of encryption gates is also expected to increase proportionally. In such cases, it is not trivial to program umpteen configurable gates. Figure 1d shows an example distribution of logical blocks that need to be programmed in a dynamically configurable security circuit. Explicit addition of auxiliary logic and peripheral circuitry is required to support dynamic programming of the configurable logic in this case. The amount of additional logic required and the resultant overhead, increase with the amount of programmability incorporated in the chip, which does not favor turning all gates reconfigurable.

In order to eliminate the dedicated auxiliary distribution logic requirements, this work proposes to integrate dynamic programmable encryption key distribution with the existing scan logic in the chip. In a typical functional unit design, logic gates are placed between flipflops, which are clocked at a designated frequency. In many systems including IBM POWER microprocessors[41,42], flipflops are designed to handle both logic and scan data. The proposed key distribution solution is shown in Fig. 1e, where scan flipflops provide encryption keybits to program the reconfigurable logic. Temporal sharing of the resources is possible since the scan programming and keybit programming do not overlap in time. Reuse and temporal sharing of the existing

scan resources obviate the need for additional complex logic programming and circuits, eliminate numerous multiplexer units required in a specialized dynamic input distribution unit, etc., thus leading to minimal perturbation in the original chip.

In summary, this work proposes a fundamentally different design scheme for logic obfuscation by having a reconfigurable active interconnect, rather than adopting poor performance polymorphic logic gates in conventional approaches, with no interference to the conventional CMOS gates. Thanks to its intrinsic transistor structure and nonvolatility, FeFET can be applied as an active interconnect. Building on this concept, the key contributions are as follow: An encryption circuit with FeFET active interconnects for functional obfuscation is proposed and experimentally demonstrated; The proposed scheme places our compact encryption logic on non-critical timing branches of the logic path, which achieves the functional obfuscation without escalating the time closure challenges; Area and energy efficient design with only 4 transistors (Previous FeFET based circuit design from Dutta et al.[17] consists a total of 28 transistors); The proposed design with active interconnects encryption blocks is highly scalable where increased encryption can be achieved by inserting more encryption blocks on the non-critical timing branches.; Our scheme makes use of the existing scan chain and scan flipflops in the chip to write into the active interconnects(Previous FeFET based circuit design[17] may require a tedious strategy and implementation for dynamic programming, as the design requires 16 different FeFETs to be programmed during runtime, whereas our design needs only 2 FeFETs to be programmed.

The rest of the article is organized as follows. Experimental verification of our proposed reconfigurable encryption block is discussed first. Details on SPICE simulation for functional verification of the circuit is followed. Our placement strategy and analysis of encryption probability on ISCAS benchmarks are discussed in subsequent sections. Peripheral programming strategies for reconfigurable blocks are discussed further. The results and conclusions are discussed in the final section. In addition, a supplementary section with additional details on device fabrication, circuit simulation, variation analysis, placement impact on encryption probability, and, block layout and analysis is also included.

## Results and discussion

**Verification of the reconfigurable block**. To verify the functionality of the proposed reconfigurable block, measurements and circuit simulations are performed. For experimental demonstration, 28 nm high-$\kappa$ metal gate FeFET devices are tested, as shown in the transmission electron microscopy cross-section images of the device Fig. 2a[38,43]. The device features a doped $HfO_2$ as the ferroelectric layer and $SiO_2$ as the interlayer in the gate stack, as shown in Fig. 2b. Detailed device information can be found in[26,38]. The FeFET memory performance is characterized by standard $I_D$–$V_G$ measurements after applying ±4 V, 1 µs write pulses on the gate. Note that a 0.1 s delay is inserted between the $I_D$–$V_G$ sweep and the memory write pulses for the trapped charges to release[27,44,45]. It is known that the charge trapping induced by write pulses counteracts the $V_{TH}$ shift caused by the polarization switching, thus reducing the memory window and degrading the endurance cycling of FeFET[27,44,45]. Inserting a delay after memory write leaves enough time for the trapped carriers to release, thus manifesting the polarization effects. Figure 2c shows a memory window about 1.2 V, i.e., the $V_{TH}$ separation between the LVT and HVT states, which enables a large ON/OFF conductance ratio. Note that, for the nucleation-limited polarization switching[46,47], a tradeoff can be realized between the write pulse amplitude and

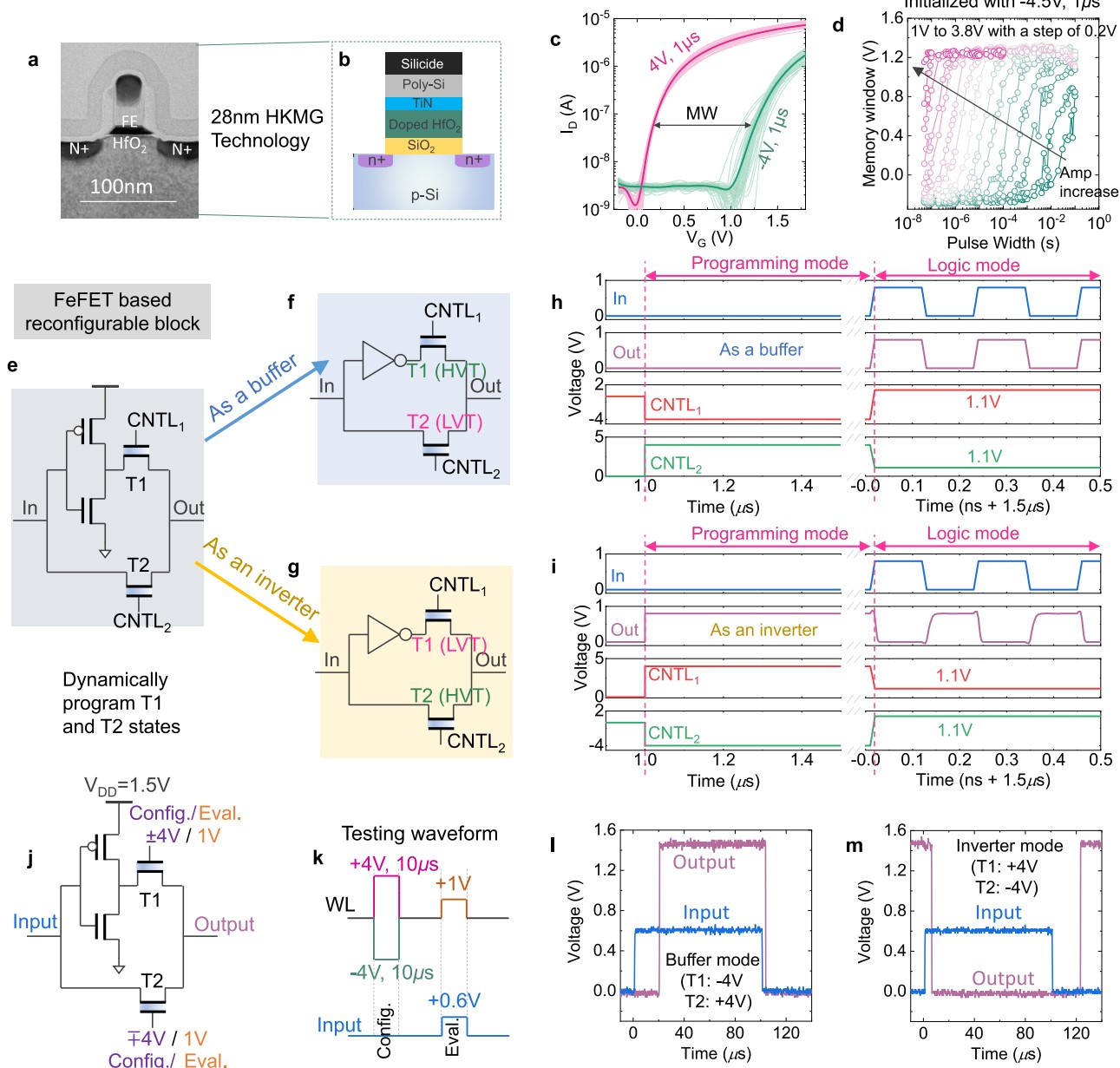

**Fig. 2 FeFET reconfigurable encryption block and its functionality verification. a** TEM cross-section[38,43] and **b** schematic cross-section of an 28 nm high-κ metal gate FeFET device. **c** $I_D$–$V_G$ characteristics of 60 different FeFETs measured after ±4 V, 1 μs write pulses. Good control over the device variability and a memory window of 1.2 V are demonstrated. **d** The dynamic switching characteristics of the FeFET as a function of write pulse width at different pulse amplitudes. Tradeoff between amplitudes and pulse widths are present. **e** Schematic of the proposed active interconnect based reconfigurable encryption block. **f** Buffer mode configuration. **g** Inverter mode configuration. **h** Simulated waveforms in buffer mode showing the programming and logic modes. **i** Simulated waveforms in inverter mode. **j** Applied voltages on the encryption block in experiment. **k** The applied waveform for functionaltiy verification in experiment. **l**, **m** Captured transient waveforms in the logic evaluation mode for buffer and inverter modes, respectively. For all tested FeFETs, $W/L = 500$ nm/500 nm are used.

pulse width, as shown in Fig. 2d, which presents the switching dynamics of FeFET as a function of applied pulse width for different pulse amplitudes. It clearly suggests that 4 V is not absolutely necessary and lower write voltages possible with a tradeoff of a large pulse width. This could help alleviate the design of peripheral supporting circuitry with lower write voltages in applications where the FeFET configuration is occasional and high speed write operation is not necessary, as the proposed logic camouflaging application in this work.

As mentioned above, the large memory window and ON/OFF conductance ratio present a unique opportunity for FeFETs to

design an active interconnect based camouflaging pass transistor (switch). In addition, the capability to dynamically shift the $V_{TH}$ makes the proposed active interconnect based FeFET-switch immune for the attacker to reverse engineer the netlist simply from layout (GDS) level. Figure 2e shows the proposed encryption block utilizing $V_{TH}$ manipulation. The proposed encryption block consists of an inverter and two FeFETs. It operates in two modes, the programming mode, and the logic mode. In programming mode, relatively high write voltages are used to program the device to set the $V_{TH}$. Once it is programmed, the device is all set to operate in the logic mode,

where a small read voltage at the gate between the $V_{TH}$ of LVT and HVT states is applied to read the FeFET state. The pass transistor either conducts or blocks the input signal based on the internal programmed $V_{TH}$ state, as shown in Fig. 2f, g, respectively. Hence an inverted input or a non-inverted input is obtained at the output of our proposed reconfigurable circuit.

The functionality of the proposed reconfigurable buffer-inverter encryption block has been verified in SPICE simulations, using a calibrated FeFET model[43] and 45 nm (NCSU FreePDK) logic transistor technology[48]. Detailed simulation parameters are shown in Table S1. In the program mode, $V_{TH}$ of the two FeFETs pass transistors are set by applying write pulses. For this study, write pulses of ±4 V, 500 ns are adopted. In the logic mode, an INPUT signal of 0.8 V at the *In* terminal and control signal of 1.1 V(read voltage chosen between the $V_{TH}$ of LVT and HVT states) at the gates of FeFETs (*CNTL1* and *CNTL2*) is asserted. In the buffer mode of encryption, as illustrated in Fig. 2f, FeFETs T1/T2 are written into HVT/LVT state respectively by asserting write voltages in CNTL1 and CNTL2 terminals, as shown in the transient waveform in Fig. 2h. In the logic (evaluation) mode, it can be seen that the output(*Out*) follows the input (*In*) as shown in Fig. 2h. On the other hand, by writing the FeFETs T1/T2 into LVT/HVT state, the inverter mode of encryption shown in Fig. 2g can be realized, which will output an inverted input signal during the logic evaluation mode, as shown in Fig. 2i. In addition, the dynamic programming of the encryption block that can switch between the buffer mode and the inverter mode has been verified in SPICE simulations, as shown in Fig. S2.

The reconfigurability of the inverter-buffer block has also been verified experimentally using the testing setup shown in Fig. 2j, k. Discrete inverter and FeFETs are assembled together for experimental verification. The relevant applied voltages are shown in Fig. 2j, k. The buffer mode and inverter mode operations are shown in Fig. 2l, m, respectively. Here only the evaluation phase waveforms are shown for clarity. Correct operations of both working modes are demonstrated. Due to the large parasitics present in the testing setup, the speed is limited to tens of μs. But it is expected that with the fully IC, high speed operations can be achieved as demonstrated in the SPICE simulations in Fig. 2h, i. SPICE analysis on threshold voltage and delay variation of the proposed encryption block is given in Fig. S3. A layout of the same is shown in Fig. S4.

To prevent RE, any camouflaging technique needs to meet two conditions[16] such as resiliency to RE and corrupted outputs. Resiliency to RE implies that an attacker will not be able to discern the functionality of the camouflaged gates. Corrupted output indicates outputs of the original netlist and deceived netlist are different. In our proposed technique, both these conditions are met. Experimental results with Fig. 2j show that the same circuit is capable of producing both the inverted and non-inverted output. Note, we apply same input voltages (both Input and Eval terminals in Fig. 2j) and the circuit gives two different outputs depending on the previously programmed (configured to either HVT or to LVT) state. This ensures that, the attacker will not be able to identify the functionality of the gate just by inspecting the physical layout. Also, the programming (configuring to LVT/HVT) of FeFETs is also done by just by applying voltages at the Config/Eval (in Fig. 2j) dynamically. The extracted netlist by the attacker will yield a different outcome, without the correct knowledge of the programmed state (buffer/inverter state) of our proposed encryption gate. Moreover, in the proposed technique, it is easy to add as many number of encryption unit (thanks to compact implementation and the easy peripheral logic) throughout the IC. This increases the functional ambiguity in the overall logic, and there by making the overall IC more immune for RE. In summary, the proposed encryption

circuit with the same input results in two different logical outputs based on the programmed states of FeFETs, making it a strong candidate for RE resilient hardware (Fig. 2h, i).

There may be several concerns related with the constant bias applied on the FeFET gates for the logic evaluation mode. First, the constant bias (+1 V in the experiment in this work) may incur significant gate leakage current, which increases the power consumption. However, this is not necessarily true for FeFETs. This is because for FeFETs, typically the ferroelectric layer thickness is rather high for a decent memory window (8 nm doped $HfO_2$ in this work), which significantly suppresses the leakage current even at 1 V, as shown in the measured gate leakage current in Fig. S5. Second concern could be stability of the FeFET states, especially under the constant bias stress for evaluation. For properly designed gate stack of $HfO_2$ FeFET, the extrapolated retention could reach 10 years at 85 °C[49,50]. We have also measured the retention of our FeFETs at different temperatures. Though it is not the best retention performance reported so far on Si FeFET, it still demonstrates window opening at 85 °C when extrapolated to 10 years, as shown in Fig. S6a. We also measured the stability of the HVT state while stressed at the evaluation gate bias (+1 V) during retention. As shown in Fig. S6b, the state is stable even at high temperature. At the end, what really determines the stability of FeFET states is the energy barrier separating the two polarization states, which can be engineered through the gate stack engineering[47]. In the last, the evaluation bias can be shifted to a lower bias value if needed through various engineering techniques, such as the channel doping or the gate work function shift[17]. Therefore, FeFET can be an excellent candidate for the proposed active interconnect based logic obfuscation application. In the next section, the encryption probability and timing beneficial placement are discussed.

**Encryption and criticality of placement**. For the proposed active interconnect based encryption blocks to function and enable resistance for RE, a judicious placement of the blocks enabling the encryption engine in a timing aware fashion is required. The location of the placement, the neighboring cells and input pattern have an impact on encryption. All these factors can contribute to logic masking effect and prevent the encrypted bit propagation to the output. In order to understand the impact of an encrypted bit and how it propagates, a circuit having a single encryption bit cell is analyzed first. Figure 3a shows an example of an encrypted circuit. C1 is the proposed buffer-inverter encryption block. Suppose $I_1$, $I_2$, $I_3$, $I_4$ to be 0, 0, 0, 1. With C1 programmed in buffer mode, the output will be a "0". However C1 in inversion mode will alter the output to be a "1". Note, if the input $I_2$ changes to bit "1" as shown in Fig. 3b, the inverted bit from C1 will not make an impact on the output, as OR gate with input 1 masks the other input. To improve the encryption strength, additional encryption key circuit can be inserted as shown in Fig. 3c. Note, 100 % inversion in the output all the time is also not reliable for security purposes as the attacker can simply resort to the negation of the output.

Timing closure is one of the most critical challenges in ASIC/ SoC designs with ever increasing clock rates[51–53]. In this work, the aim is to incorporate encryption in the IC with minimal impacts on timing critical paths. Critical paths are the longest delay paths that limit clocking. Changing the standard logic gate design to adaptable camouflaging gate design[54] for security purposes increases the critical path delay. In the proposed method, encryption gates are inserted in a non-critical timing path to overcome the potential timing failure. For example, in Fig. 3d, the path from input $I_5$ to output is the least timing critical path, as it has the minimal number of gates from the input to

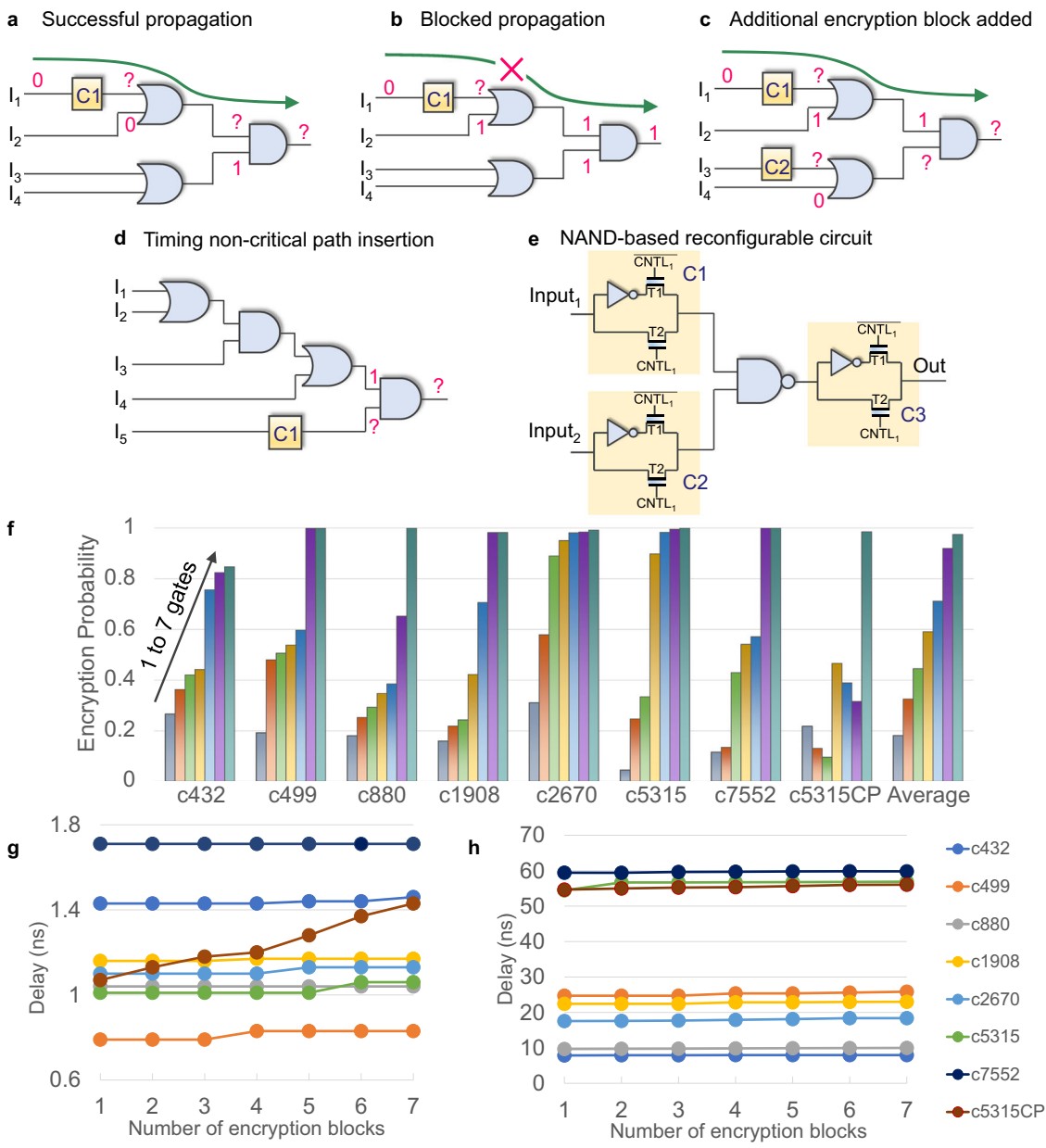

**Fig. 3 Analysis of placement ultra-compact FeFET encryption logic on encryption probability, timing and logic reconfigurability. a** Successful propagation of an encrypted bit from C1. **b** Encrypted bit from C1 blocked by OR gate. **c** Enhancing encryption probability by inserting additional encryptor. **d** Placement of interconnect based encryption unit in a non-critical timing path. **e** Reconfigurable logic based on NAND and active interconnect based encryption logic. **f** Encryption probability with increasing number of encryption blocks on ISCAS85 benchmarks (number of input test vectors: 1000). **g** Critical path delay with increasing number of encryption elements. **h** Sum of top 100 critical path delays on ISCAS circuits.

output. In this case, C1 is inserted in the logical branch from $I_5$ to the output. Though C1 is not in the timing critical path, it still logically affects output as evident from Fig. 3d.

The concept that a random placement of encryption block in the logic circuit results in different functional outputs can be extended to construct reconfigurable logic gates by systematically placing the encryption logic around standard gates. Reconfigurable logic is a known camouflaging method to obfuscate the IP. Here the attacker will not be able to discern the logic and extract the correct netlist by observing the layout. Standard cell is the building block in ICs for logic operations. The more functions it has with the same layout, the harder to be attacked. The proposed active interconnect based encryption gate can be used in conjunction with standard cells to make a very easy implementable reconfigurable logic. An example is shown with NAND gate

in Fig. 3e. Instantiating encryption logic in the inverter mode to the output of NAND gate makes the combination an "AND" gate. Adding encryption circuit to the inputs of NAND gates makes it further programmable. This makes the combination reconfigurable to NOR/OR. The internal programmed state of the instantiated interconnect based logic and potential logic gates centered around NAND gate are shown in Supplementary Material Table S2. By comparing the required number of FeFET transistors to implement the NAND gate based reconfigurable logic, shown in Table S3, the proposed encryption block is much more compact than FeFET TVD logic shown in Fig. 1a.

In this study, ISCAS85 benchmarks[55] are simulated to analyze the encryption probability. Synopsis Design Compiler is used for logic synthesis. PRIMETIME[56] is used for timing analysis. The simulations are based on the NCSU FreePDK 45 nm logic

technology[57] and a calibrated Verilog-A model of FeFET. Delay simulation of the encryption circuits is done with SPECTRE. Functional correctness is verified with test vectors using Vivado Simulator[58]. Random test vectors are generated for benchmarking. In this analysis, a non-critical path is chosen as a candidate for placement and the encryption circuit is placed randomly in the chosen path. Then test input pattern is applied to the modified circuit with encryption blocks. If the output generates an incorrect result, then output is considered to be encrypted for this input pattern. In this work, the encryption probability is defined as the fraction of times we get the incorrect output out of the total number of attempted tests.

Figure 3f shows the number of encryption element versus encryption probability for ISCAS benchmarks (c432, c499, c880, c1908, c2670, c5315 and c7552). C5315CP is the same benchmark as C5315, but with encryption blocks placed in a critical timing path and is used to show the difference in impacts compared to the proposed method. The analysis shows 84.7–100% encryption probability with an average 97.43% with a total of seven encryption blocks in the circuit. In general, the trend shows that the encryption probability increases initially with the increase in the number of encryption circuits. However, c5315CP does not show a monotonic increase in encryption probability with increase in the number of encryption blocks. For c5315CP, as mentioned earlier, encryption gates are added in the same input to the output path. This leads to double inversions in some cases and decreases the encryption probability. Here, double inversion is defined as the two time negation of data in the input-output path and a detailed example is given in Figs. S7 and S8. Also, it is observed from Fig. 3f that a comparable level of encryption probability can be achieved with a smaller number of encryption blocks for many benchmarks. For example, in C499 increasing the number of encryption blocks from 6 to 7 does not change encryption probability and it maintains at 99.9%. Similarly, also in C1908, increasing the number of encryption blocks from 6 to 7 does not increase the encryption probability from 98.2%. In C2670, increasing the number of encryption blocks from 5 to 6 increases the encryption probability only by 0.3% from 98.1 to 98.4%.

Figure 3g shows the critical timing path delay versus the added number of encryption circuits. The placement of 7 encryption blocks gives an average encryption probability of 97.43% with the increase in most critical path delay by 2.24% on average. It is observed that for most benchmarks adding encryption logic does not change the delay of the most critical path as gates are placed in different non-critical paths. For example, adding a single encryption gate does not affect the critical path delay, and adding 6 gates changes the critical path delay by only 2.04%. Note that for c5315CP, the average delay on the most critical path gets worsened by 41.58% after the insertion of 7 gates. Further, sum of delays on the top 100 critical paths is taken for each of the benchmarks to analyze the overall delay impact in the IC after the placement of our encryption blocks. Insertion of multiple encryption units on the same path can make a previously non-critical timing logic path to a critical timing path. Such occurrences are restricted by spreading out placement of encryption blocks on different logic branches. Figure 3h shows the sum of top 100 delays on ISCAS benchmarks. The impact of insertions is seen to be minimal. The placement of 5 encryption blocks gives an average encryption probability of 71.13% with an overall delay increase by 1.34%. The placement of 7 encryption blocks gives an average encryption probability of 97.43% with an overall delay increase by 3.67%.

Correlation of encryption probability with the placement of encryption logic is not linear. In the analysis, it is observed that encryption probability is at the highest if the encryption circuit is placed closer to the output node and becomes unpredictable as we move away from the output node due to logical masking. The analysis of the placement of the encryption logic in the same input-output path by varying the distance from output node is shown in Supplementary Materials Fig. S9. The analysis on ISCAS85 benchmarks demonstrates the ability to control output encryption probability without affecting the timing closure. It also shows, addition of a large number of encryption units is not necessary to get a satisfactory encryption level, which is beneficial for overall area and power savings. Typically, the placement of generic programmable gates[59] worsen the timing closure challenges, where as the proposed techniques alleviate it by restricting the placements to non-critical timing paths. Another advantage with our methodology is that instantiation of the interconnect based encryption gate from a standard cell logic library is possible on a need basis, making it easier for automation and eliminating the need for a specialized custom design of cells.

**Peripheral programming**. The proposed peripheral circuitry using scan flipflops is previously shown in Fig. 1e. Figure 4a–c shows the biasing voltage, the detailed reconfigurable encryption programming logic, and the expected waveforms for the proposed scheme. Each of the FeFET pass transistors in the key logic needs to be programmed independently in the programming phase. Once programmed, circuits operate in the logic mode of operation with logic mode voltages (i.e., FeFET state read voltages). Encryption logic is inserted along with the traditional logic circuitry as shown in Fig. 4b. As discussed earlier, complementary programming states are required for the two FeFETs to store the encryption key. Once $V_{TH}$ states are programmed into the corresponding FeFETs, a small read voltage is applied for logic operations. At first, the key sequence for programming is distributed to the scan outputs of flipflops ($S_{out}$, $\overline{S_{out}}$) through the scan chain using scan clocks ($S_{clk}$).

In the programming mode, depending on the key sequence, lower FeFET (F1) or upper FeFET (F2) get programmed to either HVT or LVT state. Scan outputs act as control signals to determine which FeFETs get to be HVT/LVT state. Typically in digital circuits, flipflop outputs (data outputs—$D_{out}/\overline{D_{out}}$, scan out—$S_{out}/\overline{S_{out}}$) are either at a logic positive voltage or at a complementary GND voltage. FeFET requires a negative voltage for its writing process to be programmed as HVT[17]. This requirement requisites the scan output to be at a negative voltage at logic low for a straightforward implementation of control biasing scheme. However, it adds complexity in the flipflop design to make the scan output voltage to be biased at negative voltage at logic zero. To avoid substantial flipflop modification by keeping scan logic zero at GND voltage and at the same time providing required write voltages to FeFETs, a two step programming process is proposed.

In the first step of programming, all FeFETs are made HVT by providing negative write voltages at L1 and L2 and keep selector transistors (T1, T2) conducting. In the second step, only the required FeFETs are converted to LVT. This is done by providing positive write voltages at L1 and L2 and NMOS based selectors (T1, T2) to be at ON/OFF state based on $S_{out}/\overline{S_{out}}$. For example if $S_{out}$ is high, it will make the T1 ON and T2 OFF, and positive write voltage will get transfered to the gate of F1(at *int1* in Fig. 4b) making the F1 LVT state. Since T2 is off, F2 will maintain the HVT state. In the logic mode, the L1 and L2 are provided with read voltages and selector transistors (T1 and T2) are turned ON by applying $V_{read}$ at L4 at L5 as well. In summary, with the proposed method, some encryption blocks invert the input logic signal and some encryption blocks buffer the output based on the provided encryption key sequence, such that overall functional obfuscation is achieved. An alternative peripheral scheme having

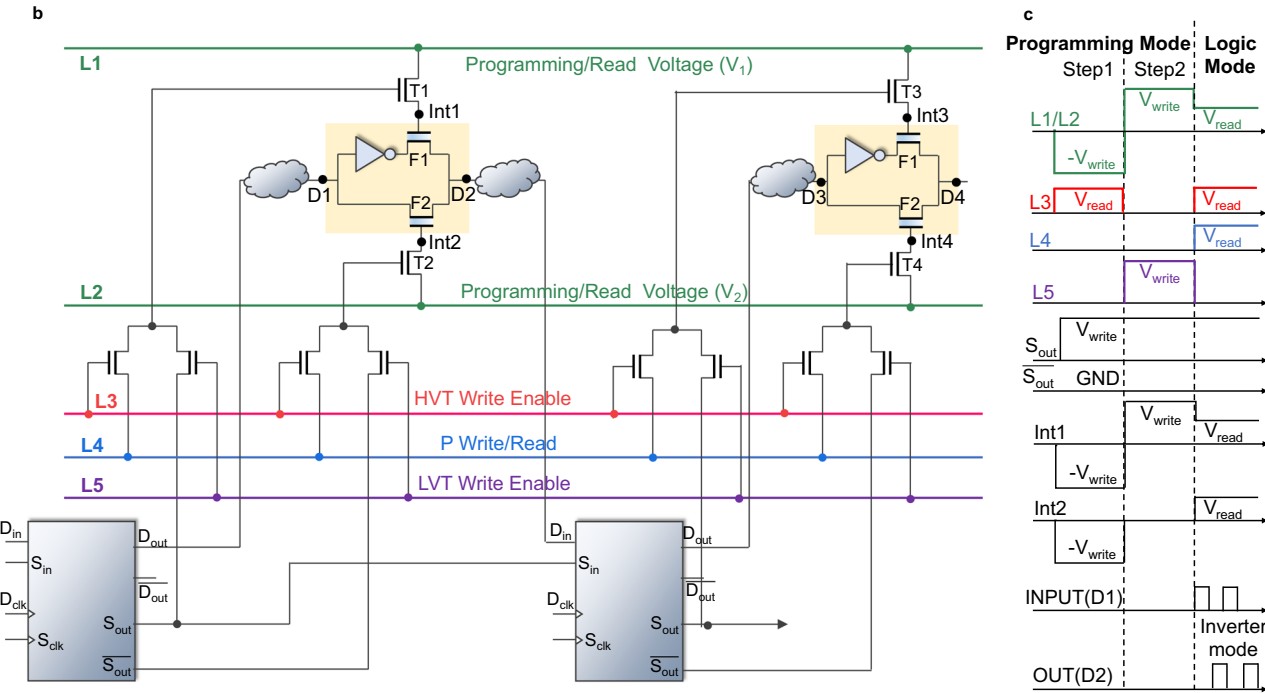

**Fig. 4 Circuit description of the proposed encryption key distribution utilizing the existing data-scan flipflops in the IC.** Gray bubbles indicates the functional logic. The circuits in the yellow enclosure show encryption logic. **a** The biasing scheme for the proposed peripheral scheme. **b** Schematic of the proposed encrypted IC with suggested peripheral circuitry by resusing the scan chain for the key distribution toward encryption blocks. **c** Peripheral biasing waveforms in the programming and logic mode operation. Note, To minimize the routing lines and periphery required, most of the metal lines are shared temporally in the operations.

one step write programming with scan flipflop output being biased at negative voltage for a logic zero is also given in the Supplementary section (Fig. S10).

In this work, ultra-compact active interconnect based on FeFET for hardware encryption is presented. FeFET leverages threshold voltage manipulation to attain run-time configurablity. The proposed encryption circuit encompassing an inverter and an active interconnect, is layout obfuscated and is capable of producing either inverted or non-inverted output. This encryption circuit is fabricated and functionality is experimentally verified. Further analysis showed that the placement of active interconnect encryption blocks in non-critical timing logic branches produces satisfactory level of encryption without jeopardizing the timing closure requirement of ICs. Analysis on ISCAS benchmark shows a 97.43% encryption probability with an average delay increase of 3.67% in the top 100 timing critical paths in ISCAS benchmarks. This work also introduced peripheral schemes for programming the reconfigurable encryption keys by reusing the scan circuity and thereby eliminating the dedicated dynamic key input distribution logic and circuitry.

## Methods

**Device fabrication**. In this paper, the fabricated FeFET features a poly-crystalline Si/TiN/doped HfO$_2$/SiO$_2$/p-Si gate stack. The devices were fabricated using a 28 nm node gate-first high-$\kappa$ metal gate CMOS process on 300 mm silicon wafers. Detailed information can be found in[26,38]. The ferroelectric gate stack process module starts with growth of a thin SiO$_2$ based interfacial layer, followed by the deposition of the doped HfO$_2$ film. A TiN metal gate electrode was deposited using

physical vapor deposition, on top of which the poly-Si gate electrode is deposited. The source and drain n+ regions were obtained by phosphorous ion implantation, which were then activated by a rapid thermal annealing at ~1000 °C. This step also results in the formation of the ferroelectric orthorhombic phase within the doped HfO$_2$. For all the devices electrically characterized, they all have the same gate length and width dimensions of 1 μm × 1 μm, respectively.

**Electrical characterization**. The FeFET device characterization was performed with a Keithley 4200-SCS semiconductor parameter analyser. Two 4225-PMUs (pulse measurement units) were utilized to make the pulsed current-voltage measurement. In the experiment, program and erase pulses were applied and the pulsed $I_D$−$V_G$ ($I_D$, drain current; $V_G$, gate voltage) measurement was performed. The total sweep duration is 5 ms. Note that, to minimize the charge trapping effects on the sensing of the programmed or erased state of the device, we inserted a delay of 100 ms between the measurement and the write pulses to allow a full trapped charge release. For the pulsed measurements, the current resolution is close to 3 nA in our set-up. The reconfigurable block characterization was performed using two FeFETs on the same chip and an externally connected inverter circuit (Texas Instruments CD74AC04E). We connected the reconfigurable block with an inverter on a breadboard. Input pulses, FeFET memory write pulses, and evaluation pulses were generated with an Keithley 4200-SCS. A 1.5 V amplitude VDD supply of the inverter was provided through an Agilent 81150A arbitrary function generator. The output voltage transient was sampled through an Tektronix TDS 2012B digital oscilloscope. All the write pulses have a pulse width of 10 μs. The input pulses have a pulse width of 100 μs and the evaluation pulse with a rising edge 5 μs ahead of the input rising edge and a falling edge 5 μs lagging behind the input falling edge. The large pulse width is chosen due to the large parasitics in our set-up. In a fully integrated reconfigurable block, the operation speed will greatly improve, as shown in the single-FeFET measurement (successful write under 20 ns, ±4 V) in Fig. 2d.

## Data availability

The data that support the plots within this paper and other findings of this study are available from the corresponding author on reasonable request.

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

## Acknowledgements

This device and circuit level analysis is supported by the U.S. Department of Energy, Office of Science, Office of Basic Energy Sciences Energy Frontier Research Centers program under Award Number DESC0021118. The architecture evaluation is supported part by the NSF under grant number 2008365 and part by ND EPSCoR. The device fabrication is funded by the German Bundesministerium für Wirtschaft (BMWI) and by the State of Saxony in the frame of the "Important Project of Common European Interest (IPCEI)".

## Author contributions

V.N., S.G., and K.N. proposed and supervised the project. Y.X. conducted the functionality verification simulation. T.Y. and Y.X. did the encryption analysis. T.Y. and N.J. did layout simulation. Y.S.K., S.D., and S.B. fabricated the FeFET devices. S.D. and Z.Z. performed the experimental characterizations of the FeFETs and encryption logic. All authors contributed to write up of the manuscript.

## Competing interests

The authors declare no competing interests.
