## [Peer Review File · Nature Communications]

REVIEWER COMMENTS

Reviewer #1 (Remarks to the Author):

The manuscript entitled “Hardware Functional Obfuscation With Ferroelectric Active Interconnects” by Yu and co-authors studies a hardware-based encryption technique. Taking advantage of ferroelectric field-effect transistor (FeFET), such an encryption technique has minimal complexity and overheads than reconfigurable logic gate implementations using several FeFETs and complex differential logic. The manuscript is well-organized and the work is of interest to scientists who are working in the field of FeFETs application. Some comments on the technical aspects of this work are listed in detail as follows:

1. As shown in Figure 2g and i, in logic mode, there are continuous voltages applied on CNTL1 and CNTL2, will it dramatically increase power consumption? The authors should also provide an energy comparison against other camouflage logic designs.
2. What is the stability of the non-volatile FeFETs? Is there drift present? How does the programmed device behave after a long duration?
3. The authors claim that for c5315CP, encryption gates added in the path lead to double inversions in some cases, which in turn decreases the encryption probability. Could the author give a more detailed explanation?
4. The authors state that “the proposed encryption circuit with the same input results in two different logical outputs based on the programmed states of FeFETs, making it a strong candidate for reverse engineering resilient hardware”, Could the author give the experimental results of its immune reverse engineering?

Reviewer #2 (Remarks to the Author):

The paper entitled “Hardware Functional Obfuscation with Ferroelectric Active Interconnects” by T. Yu et al. proposed an efficient hardware encryption technique with minimal complexity and overheads based on ferroelectric field-effect transistor (FeFET) active interconnects. The result looks interesting. However, there are several points to be addressed.

1. The FeFET device architecture in this paper is reported in ref. 40 (IEDM, 2016, 11-15) and most of the device performances are clearly presented. Moreover, Ref. 19 (IEEE Transactions on Electron

Devices 2021, 68, 516–522) has also expanded the application of the FeFET in security circuit. So the authors should clearly address the novelty of such work.

2. The device prototype is a ferroelectric field-effect transistor (FeFET) where doped HfO₂ is used as the ferroelectric layer. However, in the reported works, the interface between HfO₂ and SiO₂ should also capture charge carriers, leading to hysteresis. Such phenomenon is common in floating gate field effect transistor (such as, ACS Nano 9, 612-619, 2015; Nature Nanotechnology 12, 901–906, 2017). The authors should also address such issue in the manuscript.

3. To operate the FeFET device, one typical process is to drive the device with a definite gate voltage after program the device with the same gate terminal. If opposite voltage polarity is used (for example, the device is programmed with -4 V and operates with gate voltage of 1 V), how to ensure the stability of the polarization of the charges in the ferroelectric dielectric? Are there any data to support such results?

Hardware Functional Obfuscation With Ferroelectric Active Interconnects

Tonggunag Yu¹, Yixin Xu¹, Shan Deng², Zijian Zhao², Nicolas Jao¹,
You Sung Kim³, Stefan Duenkel³, Sven Beyer³, Kai Ni^{2*}, Sumitha George^{4*},
Vijaykrishnan Narayanan¹

¹Pennsylvania State University, State College, PA 16802, USA

²Rochester Institute of Technology, Rochester, NY 14623, USA

³GLOBALFOUNDRIES Fab1 LLC & Co. KG, Dresden, Germany

⁴North Dakota State University, Fargo, ND 58102, USA

*To whom correspondence should be addressed

Email: sumitha.george@ndsu.edu, kai.ni@rit.edu

The authors would like to thank the reviewers and editors for their constructive comments and suggestions that can significantly improve our paper. We have addressed all the comments and made all the necessary modifications to the manuscript. Detailed point-to-point response is as follows.

Review question 1.1

As shown in Figure 2g and i, in logic mode, there are continuous voltages applied on CNTL1 and CNTL2, will it dramatically increase power consumption? The authors should also provide an energy comparison against other camouflage logic designs.

The power consumed by the reconfigurable block is not significantly different from normal logic gates. Figure R1 shows the measured DC I_D-V_G and I_G-V_G on a FeFET with $W/L=500\text{nm}/500\text{nm}$. Even though a voltage is continuously applied on CNTL1 and CNTL2, but they are applied only

Figure R1: DC Transfer characteristics of a FeFET. W/L=500nm/500nm.

on the FeFET gate. Considering the relatively thick ferroelectric, typically around 10 nm, the gate leakage may even be lower than the conventional logic transistor. As shown in Figure R1, the I_G is around the noise floor of the instrument at evaluation voltage 1 V, about 100 fA. Our simulation shows around 6x difference in the energy consumption in logic mode compared to the FeFET based security design in ref. 19. This is attributed to the number of devices involved in the design. Design in ref. 19 consists 28 transistors (16 FeFETs and 12 CMOS transistors), whereas our design consists of only 4 transistors (2FeFETs and 2 CMOS transistors).

Figure R1 is added as Figure S3 in the supplementary. Discussions on the power consumption is added on page 18.

Review question 1.2

What is the stability of the non-volatile FeFETs? Is there drift present? How does the programmed device behave after a long duration?

Thanks for the reviewer for bringing this up. We have added the retention measurement on a FeFET with $W/L=500\text{nm}/500\text{nm}$ at different temperatures, as shown in Figure R2. It shows that at room temperature, extracted memory window up to 0.6 V can be retained even after 10 years, demonstrating good stability of the configured states. At higher temperatures, such as 85°C, the memory window is reduced to 0.1 V after 10 years.

Figure R2: Retention measurement of a FeFET under different ambient temperatures.

Two aspects are worth noticing. First, the presented retention data is by no means the limit of FeFET devices. Better retention in FeFETs have been reported in literature ^{1,2}. Therefore, the presented data should not be regarded as the limit of FeFET technologies and can be further

improved. Second, for the proposed camouflaging application, the required retention is much relaxed compared with the conventional nonvolatile memory, where occasional reprogramming (e.g., after months or years) can be performed.

Figure R2 is added Figure S4(a) in the supplementary. Discussion on FeFET state stability is added on page 18.

Review question 1.3

The authors claim that for c5315CP, encryption gates added in the path lead to double inversions in some cases, which in turn decreases the encryption probability. Could the author give a more detailed explanation?

For C5315CP, we conducted the experiment by adding multiple active interconnect based encryption blocks to the same input to output path. We chose to do so for analysing the impact on timing and encryption probability in such paths. We define double inversion as two times negation of data in the input-output path. Adding more than one encryption blocks(programmed in the inverter mode) to the same input-output path may lead to double inversion of data in the path for some combination of inputs. To demonstrate the logic impact of double inversion on final output we choose an input-output path with two encryption blocks. The screenshot of the timing report of the chosen input-output path with two encryption units is shown in Figure. R4. INVMOD in the timing report is the name of our encryption unit. Figure. R3a shows the schematic of a segment of the chosen input-output path (Figure. R4) before the insertion of encryption units.

Figure. R3b shows the same segment of the circuit after inserting one encryption unit in the inverter mode. Figure. R3c shows the circuit after inserting two encryption unit (both programmed to be inverters). We observe that for a set of specific inputs, the intermediate output O_{inter} is "1" as shown in Figure. R3a. After inserting a single encryption unit, O_{inter} value is switched to "0"

Figure R3: (a) A segment of input-output path. The segment output is named O_{inter} . O_{inter} is "1" with the current set of inputs. (b) Segment in (a) after adding one encryption logic(C1). C1 is programmed in the inverter mode which causes O_{inter} to be "0". (c) Segment in (a) after adding two encryption logic units(C1 & C2). Both C1 & C2 are programmed in the inverter mode which makes O_{inter} to be "1" again. (d) Rest of the schematic from the intermediate output to the selected path's final output. It is seen that changes in intermediate output O_{inter} (due to double inversion) affects the outcome of final output with certain input combinations.

```

Report : timing
-path_type full
-delay_type max
-slack_lesser_than 0.00
-max_paths 200
-transition_time
-capacitance
-sort_by slack
Design : c5315mod
Version : K-2015.12-SP2
Date : Fri May 28 05:12:35 2021
*****

```

```

Startpoint: in[75] (input port)
Endpoint: out[0] (output port)
Path Group: (none)
Path Type: max

```

Point	Cap	Trans	Incr	Path
input external delay			0.00	0.00 f
in[75] (in)	0.02	0.00	0.00	0.00 f
U1536/Y (MUX2X1)	0.01	0.09	0.07	0.07 r
U1535/Y (INVX1)	0.00	0.01	0.04	0.12 f
U1078/Y (AND2X1)	0.01	0.03	0.05	0.17 f
U1079/Y (INVX1)	0.00	0.00	0.01	0.18 r
U1534/Y (AOI21X1)	0.00	0.01	0.01	0.19 f
U1533/Y (INVX1)	0.03	0.12	0.09	0.29 r
U1044/Y (AND2X1)	0.01	0.05	0.05	0.33 r
U1045/Y (INVX1)	0.00	0.01	0.03	0.36 f
U1513/Y (AOI21X1)	0.00	0.05	0.04	0.40 r
inverter1/Y (INVMOD)	0.01	0.07	0.07	0.48 f
U1508/Y (AOI21X1)	0.00	0.02	0.03	0.51 r
U988/Y (BUF2)	0.00	0.01	0.04	0.55 r
inverter2/Y (INVMOD)	0.00	0.05	0.06	0.61 f
U1507/Y (XOR2X1)	0.01	0.09	0.07	0.68 r
U1502/Y5 (FAX1)	0.00	0.01	0.09	0.78 f
U1501/Y (MUX2X1)	0.01	0.08	0.07	0.84 r
U1500/Y (XOR2X1)	0.00	0.02	0.04	0.89 f
U1467/Y (AOI22X1)	0.00	0.04	0.04	0.93 r
U840/Y (BUF2)	0.00	0.01	0.04	0.96 r
U808/Y (AND2X1)	0.00	0.03	0.03	0.99 r
U963/Y (INVX1)	0.01	0.04	0.04	1.03 f
U1466/Y (MUX2X1)	0.00	0.04	0.05	1.08 r
U1465/Y (MUX2X1)	0.00	0.01	0.03	1.11 f
U1464/Y (NAND2X1)	0.00	0.03	0.02	1.13 r
out[0] (out)		0.03	0.00	1.13 r
data arrival time				1.13

(Path is unconstrained)

O_{inter} →

Figure R4: Timing report of a path from c5315CP

as shown in Figure. R3b. After inserting two encryption units , O_{inter} value is switched back to "1" as shown in Figure. R3c (double inversion at O_{inter}). Figure. R3d shows that changes in intermediate output (O_{inter}) gets transmitted to the final output with certain input combinations. This implies that more than one encryption units in path may leave the original output unchanged in certain conditions decreasing the encryption probability.

Figure R4 is added as Figure S8 and Figure R3 is added Figure S9 in the supplementary. Relevant texts are added into the last paragraph on page 22.

Review question 1.4

The authors state that “the proposed encryption circuit with the same input results in two different logical outputs based on the programmed states of FeFETs, making it a strong candidate for reverse engineering resilient hardware”, Could the author give the experimental results of its immune reverse engineering?

For reverse engineering, hackers tried to extract circuit netlist (functional information) from physical layout. IC reverse engineering involves steps such as depackaging, delayering, high resolution imaging and annotation for extracting gate level netlists³. To prevent reverse engineering, any camouflaging technique needs to meet two conditions³ such as resiliency to reverse engineering and corrupted outputs. Resiliency to reverse engineering indicates that an attacker will not be able to discern the functionality of the camouflaged gates. Corrupted output indicates outputs of the original netlist and deceived netlist are different. In our proposed technique both these con-

ditions are met. Our experimental results shows that the same circuit (Fig.2(j) from the paper) produces both the inverted (Fig.2(m))and non inverted output(Fig.2(l)). Note we apply the same input voltages (both Input and Eval terminals in Fig. 2j) and the circuit gives two different outputs depending on the previously programmed(configured to either HVT or to LVT) state. This ensures that, the attacker will not be able to identify the functionality of the gate just by inspecting the physical layout. Note, the programming (configuring to LVT/HVT) is also done by just by applying voltages at the Config/Eval(in Fig. 2j) dynamically. The extracted netlist by the attacker will yield a different outcome, without the correct knowledge of the buffer/inverter state of our proposed encryption gate. Moreover, in our proposed technique, it is easy to add as many number of encryption units (thanks to compact implementation and the easy peripheral logic) throughout the IC. This increases the functional ambiguity in the overall logic, and there by making the overall IC more immune for reverse engineering.

Discussions on reverse engineering are added on page 17.

Review question 2.1

The FeFET device architecture in this paper is reported in ref. 40 (IEDM, 2016, 11-15) and most of the device performances are clearly presented. Moreover, Ref. 19 (IEEE Transactions on Electron Devices 2021, 68, 516–522) has also expanded the application of the FeFET in security circuit. So the authors should clearly address the novelty of such work.

Thanks for the reviewer bringing this point up. The key novelties of this work lie not in the

new FeFET device, but in how we leverage the FeFET to realize the logic obfuscation. Typical digital circuits are composed of logic gates and interconnect. Most of logic obfuscation work focus on developing polymorphic logic gates, which can realize different logic functions depending on the configuration. One example is ref. 19, which leverages the reconfigurable threshold voltage states of FeFET and complex sensing circuitry to support the logic function (in total 16 FeFETs and 12 CMOS transistors). In addition to the area penalty, most of the times, these logic gates are constructed with devices not optimized for logic operations, which causes degradation in latency. Such logic gates mixing together with the normal CMOS logic gates create significant challenge for satisfying timing closure constraints.

This work proposes a fundamentally different design scheme for logic obfuscation by having a reconfigurable active interconnect, with no interference to the conventional CMOS gates. Thanks to its intrinsic transistor structure and nonvolatility, FeFET can be applied as an active interconnect. Building on this concept, our key contributions are as follow: 1) We propose an encryption circuit with FeFET active interconnects for functional obfuscation and experimentally demonstrate it; 2) We propose to place our compact encryption logic on non-critical timing branches of the logic path, which achieves the functional obfuscation without escalating the time closure challenges; 3) Area and energy efficient design with only 4 transistors (ref 19 contains a total of 28 transistors); 4) Our design with active interconnects encryption blocks is highly scalable where increased encryption can be achieved by inserting more encryption blocks on the non-critical timing branches.; 5) We make use of the existing scan chain and scan flip-flops in the chip to write into the active interconnects. Distributing the key bits to all the encryption logic units through temporal reuse

of scan chain in the chip, obviates the need of a large dedicated key writing circuitry that would span across the entire chip. Design described in Ref. 19 design may require a tedious strategy and implementation for dynamic programming, as the design requires 16 different FeFETs to be programmed during runtime.

The relevant discussions are added into the second last paragraph of the Introduction section on page 11

Review question 2.2

The device prototype is a ferroelectric field-effect transistor (FeFET) where doped HfO₂ is used as the ferroelectric layer. However, in the reported works, the interface between HfO₂ and SiO₂ should also capture charge carriers, leading to hysteresis. Such phenomenon is common in floating gate field effect transistor (such as, ACS Nano 9, 612-619, 2015; Nature Nanotechnology 12, 901–906, 2017). The authors should also address such issue in the manuscript.

The authors thank the reviewer for pointing this out. Indeed charge trapping is a serious concern for HfO₂ based FeFET. It is known that charge trapping has an opposite effect on the FeFET compared with polarization switching, reducing the memory window⁴⁻⁶

To minimize the impact of charge trapping induced by write pulses on the FeFET operation, we insert a 0.1s delay between the memory read and memory write such that trapped carriers can be released during the delay. In this way, the charge trapping impact on the FeFET characteristics

can be minimized.

Relevant discussions are added into the first paragraph of Section "Verification of the Reconfigurable Block" on page 12.

Review question 2.3

To operate the FeFET device, one typical process is to drive the device with a definite gate voltage after program the device with the same gate terminal. If opposite voltage polarity is used (for example, the device is programmed with -4 V and operates with gate voltage of 1 V), how to ensure the stability of the polarization of the charges in the ferroelectric dielectric? Are there any data to support such results?

We have measured the stability of FeFET reset state, i.e., after -4 V write, when subjecting to 1 V stress during retention, as shown in Figure R5. The V_{TH} of the reset state shows good stability under different temperatures. This result show that leveraging the FeFET devices shown in this work could enable the proposed hardware camouflaging without significant issue from the state stability.

On the other hand, if the state is not stable under constant stress, there are also strategies to solve this issue. The V_{TH} engineering methods, such as gate metal work function and interfacial dipole engineering, can be leveraged to shift the evaluation gate bias to lower value to minimize the disturb. There is no fundamental roadblock of keep applying a small gate bias for evaluation

purpose.

Figure R2 is added Figure S4(b) in the supplementary. Discussion on FeFET state stability under a small voltage bias is added on page 18.

Figure R5: FeFET Retention when an 1 V gate voltage is constantly applied after reset with -4V.

References

1. Müller, J. *et al.* Ferroelectric hafnium oxide: A cmos-compatible and highly scalable approach to future ferroelectric memories. In *2013 IEEE International Electron Devices Meeting*, 10–8 (IEEE, 2013).
2. Beyer, S. *et al.* Fefet: A versatile cmos compatible device with game-changing potential. In *2020 IEEE International Memory Workshop (IMW)*, 1–4 (IEEE, 2020).

3. Rajendran, J., Sam, M., Sinanoglu, O. & Karri, R. Security analysis of integrated circuit camouflaging. In *Proceedings of the 2013 ACM SIGSAC Conference on Computer & Communications Security, CCS '13*, 709–720 (Association for Computing Machinery, New York, NY, USA, 2013). URL <https://doi.org/10.1145/2508859.2516656>.
4. Yurchuk, E. *et al.* Charge-trapping phenomena in hfo 2-based fefet-type nonvolatile memories. *IEEE Transactions on Electron Devices* **63**, 3501–3507 (2016).
5. Gong, N. & Ma, T.-P. A study of endurance issues in hfo 2-based ferroelectric field effect transistors: Charge trapping and trap generation. *IEEE Electron Device Letters* **39**, 15–18 (2017).
6. Ni, K. *et al.* Critical role of interlayer in hf 0.5 zr 0.5 o 2 ferroelectric fet nonvolatile memory performance. *IEEE Transactions on Electron Devices* **65**, 2461–2469 (2018).

REVIEWER COMMENTS

Reviewer #1 (Remarks to the Author):

The authors have addressed most of the questions raised by the reviewers. Another question is whether the floating gate transistor can replace the ferroelectric transistor in this design. The authors are suggested to compare the two devices.

Reviewer #2 (Remarks to the Author):

The authors have well adressed all my questions, and I am happy to adress the publication of the work.

Hardware Functional Obfuscation With Ferroelectric Active Interconnects

Tonggunag Yu¹, Yixin Xu¹, Shan Deng², Zijian Zhao², Nicolas Jao¹,
You Sung Kim³, Stefan Duenkel³, Sven Beyer³, Kai Ni^{2*}, Sumitha George^{4*},
Vijaykrishnan Narayanan¹

¹Pennsylvania State University, State College, PA 16802, USA

²Rochester Institute of Technology, Rochester, NY 14623, USA

³GLOBALFOUNDRIES Fab1 LLC & Co. KG, Dresden, Germany

⁴North Dakota State University, Fargo, ND 58102, USA

*To whom correspondence should be addressed

Email: sumitha.george@ndsu.edu, kai.ni@rit.edu

The authors would like to thank the reviewers and editors for going over our revised manuscripts. For the additional concern, we have modified our manuscript and provided a point-to-point response here.

Review question 1.1

The authors have addressed most of the questions raised by the reviewers. Another question is whether the floating gate transistor can replace the ferroelectric transistor in this design. The authors are suggested to compare the two devices.

Yes, in principle the single transistor floating gate memory could also be applied in this case. However, integrating the embedded flash memory and distributed them randomly throughout the chip is very expensive. It has been known that scaling the embedded flash memory down to 28 nm and below is challenging due to the thick gate stack (~20 nm) and added cost due to additional

masks ¹. However, integration of FeFET is much more easier and has much better scalability. Demonstration of FeFET on advanced transistor technologies, such as the FinFET ² and gate-all-around transistor ^{3,4}, have been reported. Therefore, FeFET shows much better scalability and ease of integration compared with the embedded flash memory. On the other hand, from the electrical performance perspective, FeFET also excels compared with the flash memory. Due to its thick gate stack, embedded flash generally need a large write voltage and has a slow write speed. FeFET, however, can work with below 4 V and 10 ns write pulses, demonstrating superior write performance. Therefore, flash memory could, in principle, work in our design, but with more significant challenges.

Relevant texts are added into first paragraph on page 8 and page 9. New references 19, 31-34 are added.

References

1. Strenz, R. Review and outlook on embedded nvm technologies—from evolution to revolution. In *2020 IEEE International Memory Workshop (IMW)*, 1–4 (IEEE, 2020).
2. Yan, S.-C. *et al.* High speed and large memory window ferroelectric hfzro finfet for high-density nonvolatile memory. *IEEE Electron Device Letters* **42**, 1307–1310 (2021).
3. Huang, W. *et al.* Ferroelectric vertical gate-all-around field-effect-transistors with high speed, high density, and large memory window. *IEEE Electron Device Letters* **43**, 25–28 (2021).

4. Lee, S.-Y., Lee, C.-C., Kuo, Y.-S., Li, S.-W. & Chao, T.-S. Ultrathin sub-5-nm hf zro for a stacked gate-all-around nanowire ferroelectric fet with internal metal gate. *IEEE Journal of the Electron Devices Society* **9**, 236–241 (2021).

REVIEWERS' COMMENTS

Reviewer #1 (Remarks to the Author):

The authors have made satisfactory revisions. I would recommend the acceptance.